

# Introducing ELSA v2.0: an isochronal model for ice-sheet layer tracing

Therese Rieckh[1,2], Andreas Born[1,2], Alexander Robinson[3], Robert Law[1,2], and Gerrit Gülle[4]

[1]University of Bergen, Department of Earth Science, Bergen, Norway
[2]Bjerknes Centre for Climate Research, Bergen, Norway
[3]Alfred Wegener Institute, Helmholtz Centre for Polar and Marine Research, Potsdam, Germany
[4]University of Hamburg, Department of Theoretical Physics, Hamburg, Germany

**Correspondence:** Therese Rieckh (therese.rieckh@uib.no)

**Abstract.** We provide a detailed description of the ice-sheet layer age tracer ELSA — a model that uses a straightforward method to simulate the englacial stratification of large ice sheets — as an alternative to Eulerian or Lagrangian tracer schemes. ELSA's vertical axis is time and individual layers of accumulation are modeled explicitly and are isochronal. ELSA is not a stand-alone ice-sheet model, but requires uni-directional coupling to another model providing ice physics and dynamics (the "host model"). Via ELSA's layer tracing, the host model's output can be evaluated throughout the interior using ice core or radiostratigraphy data. We describe the stability and resolution-dependence of this coupled modeling system using simulations of the last glacial cycle of the Greenland ice sheet using one specific host model. Key questions concern ELSA's design to maximize usability, which includes making it computationally efficient enough for ensemble runs, as well as exploring the requirements for offline forcing of ELSA with output from a range of existing ice-sheet models. ELSA is an open source and collaborative project, and this work provides the foundation for a well-documented, flexible, and easily adaptable model code to effectively force ELSA with (any) existing full ice-sheet model via a clear interface.

## 1 Introduction

Large ice sheets preserve climate-cryosphere interactions of the past, such as accumulation or melt rates, which result in isochronal layers (i.e. layers of same age) with particular characteristics in the ice sheets' interiors. These records are accessible via e.g. ice cores, which have been used extensively to study the climate of the past and climate-ice interactions (e.g., North Greenland Ice Core Project members, 2004; Dahl-Jensen et al., 2013). However, ice core data are limited to specific locations and are sparsely available due to their high cost and time consumption. Radar observations, on the other hand, provide three dimensional englacial stratigraphy data of the Greenland and Antarctic ice sheets (MacGregor et al., 2015; Cavitte et al., 2021). While further data acquisition and processing is required for the Antarctic ice sheet (AntArchitecture, 2017), isochrones have been traced and dated over the majority of the Greenland ice sheet. This high-quality data set, although currently under-exploited, is the ideal tool to evaluate ice-sheet model output with observations throughout the ice sheet's interior, provided that the model used features an age tracer.





But how can the englacial stratigraphy be modeled accurately? Standard tools are Eulerian and Lagrangian tracer schemes. However, the Eulerian approach suffers from instability and numerical diffusion effects while in the Lagrangian approach
dispersion of the discrete trace particles causes errors and loss of information with depth (Rybak and Huybrechts, 2003). Semi-Lagrangian schemes — where new particles are spawned at every time step, thereby avoiding some of the drawbacks of regular Lagrangian schemes — have been increasingly used for age tracing with good results (Clarke and Marshall, 2002; Clarke et al., 2005; Lhomme et al., 2005; Goelles et al., 2014). However, semi-Lagrangian schemes require perpetual three-dimensional interpolation to determine the values at the location of newly created particles, which is costly and may deteriorate
the solution over time.

More recently, Born (2017) developed an alternative, straightforward approach: an isochronal model where the englacial stratigraphy is modeled explicitly with individual layers of accumulation, driven by surface mass balance. Each layer added is isochronal, i.e. has a fixed time stamp. As time passes and more layers accumulate, ice of older layers flows towards the margin of the ice sheet and the layers become thinner. This approach faithfully represents the englacial stratification and
eliminates unwanted diffusion in the vertical direction as layers never exchange mass. In Born (2017), the isochronal model is a 2-dimensional ice-sheet model including ice flow description and parametrization. Born and Robinson (2021) isolated the layer tracer scheme into a separate module and coupled it to the full ice-sheet model Yelmo (described in Robinson et al. (2020)), which provides the necessary parameters for the layer tracing. Born and Robinson (2021) then applied this isochronal model for the Greenland ice sheet over the last glacial cycle, demonstrating that it produces a more reliable simulation of the
englacial age profile than Eulerian age tracers.

Here we provide the full description of this isochronal model ELSA (Englacial Layer Simulation Architecture) version 2.0 as an independent tool and with an improved layer accumulation scheme and more parameter choice for flexibility. The work flow of the model as well as key features are presented in Section 2. Section 3 provides a description of the control run with the coupled ELSA-Yelmo setup and the reference radiostratigraphy data set. Section 4 contains the description and results of
running ELSA with decreased resolution to decrease its computational requirements, which enables large ensemble runs for ice-sheet model parameter tuning. Section 5 provides a summary and conclusion.

## 2  Model design and description

ELSA is not a stand-alone ice-sheet model, but requires uni-directional coupling to a full ice-sheet model (the "host model") providing the boundary conditions related to ice physics (surface mass balance SMB, basal mass balance BMB, ice thickness)
and ice dynamics (horizontal ice velocities). This setup makes ELSA's layer tracing feature widely applicable, as ultimately *any* ice-sheet model can be used as host model.

### 2.1  Key features

– ELSA **explicitly represents individual layers of accumulation**, each of which has a fixed timestamp (i.e., each layer is isochronal). Horizontal flow is Eulerian and exclusive to each layer, while vertical change is Lagrangian. Layers never





exchange mass. This approach avoids advection across the vertical grid interfaces and eliminates numerical diffusion by design.

–   ELSA has a **vertical grid that is defined in time rather than space** (Born, 2017). The vertical resolution of ELSA is given by how frequently a new layer is added. The isochronal scheme therefore defines depositional age as the grid and layer thickness as an advected property, opposite from a Eulerian scheme. Note that layers are not required to be

equidistant in time or space.

–   ELSA **simulates its own layered ice sheet independently from the host model and based entirely on the ice physics and dynamics of the host model**. ELSA does not alter the simulation of the host model in any way, it simply adds the feature of tracing the age and thickness of the individual layers comprising the ice sheet. ELSA's ice sheet evolves on the host model's horizontal grid with the same temporal resolution as the host model's.

–   **It is possible to force ELSA offline** by providing input data to ELSA after the host model has completed its simulation. A prerequisite for offline forcing is that the required input data is stored with sufficiently high temporal resolution.

–   **ELSA's resolution can be controlled** with the following parameters: 1) setting the vertical resolution via the parameter *layer_resolution* for a regular grid or *age_elsa_iso* for customized isochrones; 2) reducing the horizontal resolution with respect to the host model via the parameter *grid_factor*; 3) controlling the coupling period (*CP*) between the host model

and ELSA via the parameter *update_factor*, i.e. setting ELSA's temporal resolution. Table 1 provides a description of all of ELSA's parameters.

–   ELSA is currently available in **Fortran 2003 and it is object oriented**. Its only dependency is the Library of Iterative Solvers for linear systems (https://www.ssisc.org/lis/, last access: Mar 24, 2023) (Nishida, 2010), which is used for solving the advection equation.

–   ELSA's code is **open source** and published under the GNU General Public License v3 at https://git.app.uib.no/melt-team-bergen/elsa.

## 2.2   Design and work flow

The interface between ELSA and the host model is concise and well defined. Over the entire simulation, ELSA requires only surface and basal mass balances, 3D horizontal velocity fields, and ice thickness from the host model. The workflow of the coupled model setup is depicted in Fig. 1 and executes as follows:

At the beginning of the simulation, ELSA's ice sheet is initialized with 10 equally thick layers (initialization layers). These are typically much thicker than the layers added to the ice sheet during the simulation and do not represent specific isochrones of interest.

After the host model's equilibration and spin-up, the regular model run starts with its defined time step size *dt*. ELSA can be updated at every time step of the host model (update factor *UF*=1, coupling period *CP*=*dt*) or less frequent (*UF*>1,

$CP = UF \cdot dt$). Whenever ELSA is updated using the host model's parameters, the following happens:





**Table 1.** ELSA's parameters

| parameter | default | description |
| --- | --- | --- |
| n_layers_init | 10 | number of initialization layers in ELSA |
| elsa_out | 2000 a | ELSA's output time step |
| use_dye_tracer | False | flag to set dye tracer |
| allow_pos_bmb | False | allow positive BMB (added to bottom layer) |
| layer_resolution ($LR$) | 200 a | vertical resolution of ELSA |
| age_elsa_iso ($AEI$) | None | set the ages of the desired isochrones directly (a Common Era) |
| grid_factor ($GF$) | 1 | number of host model grid boxes to be averaged over in one dimension to compute one ELSA grid box |
| update_factor ($UF$) | 1 | factor to control the coupling period, $CP = dt \cdot UF$, where $dt$ is the host model's time step |

1. Using the host model's SMB and BMB, the thickness of the top and bottom layers is adjusted. Positive SMB represents snow accumulation and is added to the top layer. Negative SMB represents melt and the top layer is thinned accordingly. If the top layer is thinner than the amount of melt, the top layer's thickness is reduced to zero and the layer(s) below are thinned accordingly. The host model's BMB is used to adjust the thickness of the bottom layer exclusively (which is an intialization layer). Negative BMB represents melt and the bottom layer is thinned accordingly. If the host model allows positive BMB, ELSA's flag *allow_pos_bmb* controls if the bottom layer can gain thickness.

2. ELSA's individual layers are advected in the horizontal dimension using a Eulerian description of flow (see Appendix B). The passive tracer variable is the layer thickness $d$, which is advected using an implicit upstream scheme and the host model's horizontal velocities, which are linearly interpolated in the vertical onto the isochronal grid. All layers are advected, where advection is strictly two-dimensional within each isochrone and mass is never exchanged among layers.

3. ELSA's ice sheet thickness is normalized to the host model ice sheet thickness to avoid drift away from the host model state.

At the end of the ELSA update, a new layer may be added on top of the ice sheet depending on ELSA's vertical resolution, which is either determined by the layer resolution (regular grid) or by manually setting isochrone timestamps. The new top layer is then filled according to the host model's SMB until the next layer is added.

The simulation returns to the host model, where output may be written to file, before the host model proceeds to the next time step to continue the simulation.

In summary, fluxes of mass from the host model are used to dynamically adjust ELSA's top and bottom layer thickness, host model horizontal velocities are used to advect ELSA's layers, and host model ice sheet thickness is used to normalize ELSA's ice sheet to avoid drift away from the host model state.





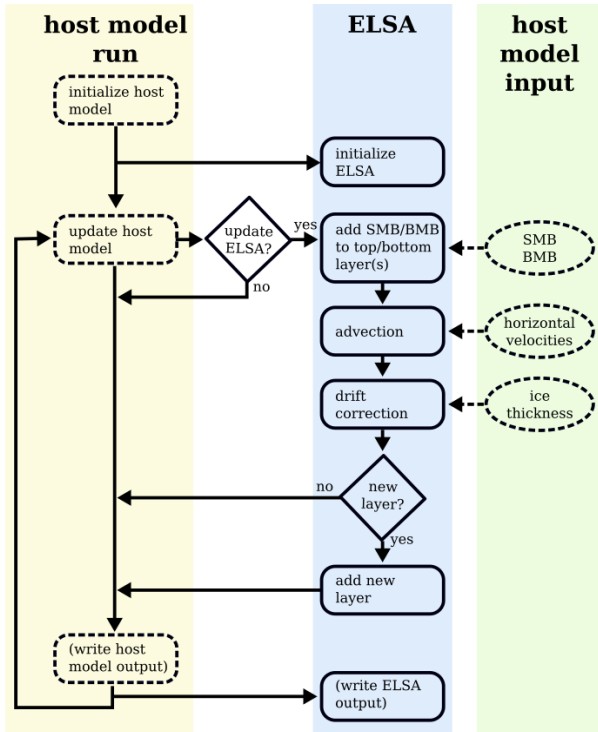

**Figure 1.** ELSA's work flow and interaction with the host model. Square dashed boxes describe host model actions, square solid boxes ELSA actions. Oval dashed boxes mark parameters required as input from the host model to ELSA. Diamond-shaped boxes mark decision points in ELSA on temporal and vertical resolution (see Sec. 2.2.1 for details).

### 2.2.1 Lowering ELSA's resolution

In the default setup, ELSA solves the advection equation for every layer at every host model grid point and time step, which is computationally costly. Here we describe various ways to decrease the computational cost of ELSA by reducing the spatiotemporal resolution of input data, which is also important to assess the feasibility of offline simulations. ELSA's spatio-temporal

110 resolution is controlled by the following parameters:

  The model parameter *layer_resolution* (*LR*) adjusts the vertical resolution in ELSA (how often a new layer is added) for regular vertical grids. Alternatively, desired isochrones can be set directly with the model parameter *age_elsa_iso* (*AEI*). A lower vertical resolution means that fewer, but thicker, layers are advected at every time step. A lower vertical resolution may be acceptable in cases where only a few or selected isochrones are of interest, for example in comparisons with reconstructions.

115   The model parameter *grid_factor* (*GF*) changes the horizontal resolution in ELSA without changing the horizontal resolution of the host model. If set to a value greater than 1, *GF* number of host model grid boxes are averaged over in both x and y direction to one ELSA grid box. E.g. for *GF*=3, the average is computed over a total of 9 host model grid boxes to determine




1 ELSA grid box. Values of *GF*>1 are useful when the horizontal resolution of the host model is much higher than that of, e.g., a data set of reconstructions. Note that *GF* has to be an integer ≥1.

The model parameter *update_factor* (*UF*) controls the coupling period $CP = UF \cdot dt$ between the host model and ELSA, i.e. how often ELSA is updated with the host model's ice parameters. The effect of less frequent coupling is twofold and impacts both accumulated mass balance and advection. The most recent host model SMB and BMB, scaled in time with *CP*, are used to update the thickness of ELSA's top and bottom layers. In the same manner, the most recent host model horizontal velocities are used to advect all layers in ELSA. While *UF*>1 decreases run time, it also demonstrates the effect of tracing layers with

less information, as would be the case in an offline application of ELSA, fed with less output saved from an ice-sheet model. Note that *UF* has to be an integer ≥1.

## 3    Control run setup and results

We run ELSA online coupled with the open source thermomechanical ice-sheet model Yelmo. Yelmo solves for the coupled velocity and temperature solutions of the ice sheet (Robinson et al., 2020), where the ice dynamics are solved with a depth-

integrated-viscosity approximation approach (Robinson et al., 2022). The model uses Glen's flow law with an exponent of 3. The horizontal grid is an evenly spaced Cartesian grid. We chose to run simulations at a resolution of $16\,\mathrm{km}$ as a compromise between speed and output detail for the experiments. The vertical grid is a sigma-coordinate grid (Greve and Blatter, 2009) with 10 layers, which are of higher resolution at the base of the ice sheet. The geothermal heat flux field is prescribed using Shapiro and Ritzwoller (2004). Bedrock and ice topography are taken from Morlighem et al. (2017). The present-day climate

is from Fettweis et al. (2017). The past climate is determined transiently using a glacial index method (e.g., Tabone et al., 2018), with the climate interpolated between present day and a Last Glacial Maximum (LGM) climate field. The LGM climate field is defined as $2\,°\mathrm{C}$ colder than the climatological average of several global circulation models from the PMIP3 project (Kageyama et al., 2021), as basal ice temperatures were consistently too warm without the $2\,°\mathrm{C}$ adjustment. Surface mass balance is computed using the positive-degree-day (PDD) scheme with a snow melt factor of $3\,\mathrm{mm\,K^{-1}\,d^{-1}}$ water equivalent

and ice melt factor of $8\,\mathrm{mm\,K^{-1}\,d^{-1}}$ water equivalent. Basal friction is exponentially scaled with bedrock elevation.

We used a 700 member ensemble to tune Yelmo parameters and evaluated Eemian ice extent, LGM ice extent, present-day ice extent and thickness, present-day surface velocities, and ice bed properties (frozen versus thawed) to find a suitable Yelmo control (CTRL) run (see Fig. A1). The overall development of ice sheet area and volume over the last glacial cycle agrees well enough with previous estimates (e.g., Vasskog et al., 2015) and the present-day bed properties are similar to the ones estimated

by MacGregor et al. (2022).

Yelmo can employ an internal adaptive timestepping scheme, but is controlled in our experiments by an outer time loop set with a value of $dt = 10\,\mathrm{a}$. ELSA's parameter *UF* is set to 1 for this simulation, meaning that advection is updated every 10 years. Simulations are run from $160\,\mathrm{ka}$ BC until the year 2000 AD with ELSA's default vertical (layer) resolution of $200\,\mathrm{a}$. Thus the isochronal grid contains a total of 820 layers (10 initialization layers and 810 isochrones) at the end of the simulation.

The horizontal grid of the layer tracing scheme is the same as in Yelmo.





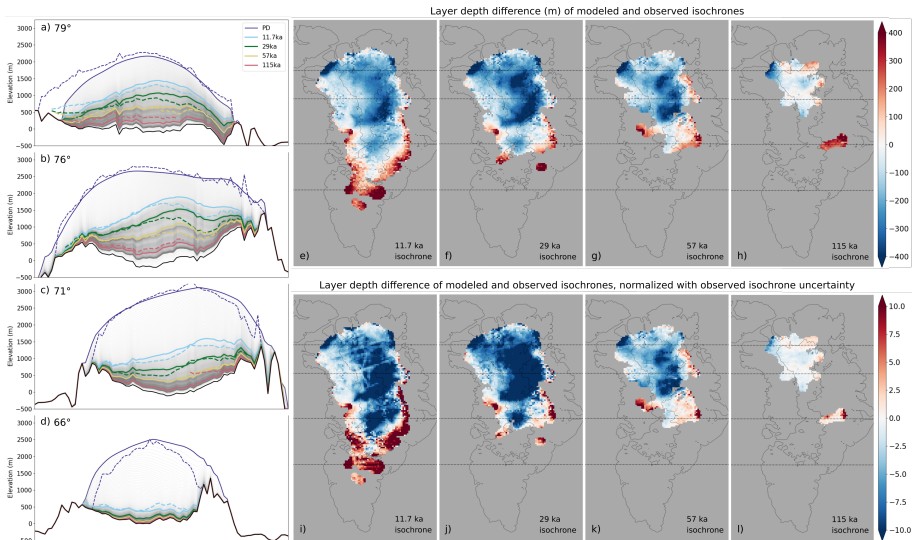

**Figure 2.** Layer depth differences between modeled and observed isochrones: a)–d) cross sections of the Greenland ice sheet at four different latitudes. Colored dashed lines mark the OIB ice sheet surface and four isochrones, the equivalent modeled isochrones of same age are colored accordingly. Thin grey lines mark ELSA's isochrones every 200 a. The thin dotted lines in panels e)–l) mark the approximate latitudes of the cross sections. e)–h) layer depth difference (m) between modeled and observed isochrones. i)–l) layer depth difference normalized with observed isochrone uncertainty (according to MacGregor et al. (2015)) between modeled and observed isochrones.

Isochrones derived from the radiostratigraphy data of the Operation IceBridge (OIB) project (MacGregor et al., 2015) can be used to evaluate ELSA's output from the coupled CTRL run. The OIB isochrone dating is based on Greenland Ice Core Chronology 2005 extended time scale, where age is defined as the year before 2000 a AD. For the remainder of the manuscript, dates and ages with the unit ka are referring to ka before 2000 a AD.

MacGregor et al. (2015) provide depth below surface and layer depth uncertainty of four dated isochrones of the Greenland ice sheet, as well as ice thickness. The ages of the four OIB isochrones are 11.7 ka, 29 ka, 57 ka, and 115 ka. Isochrone depth uncertainty is a combination of true depth uncertainty for traced reflections (about 10 m), age uncertainty of the ice core data at the appropriate depth, age uncertainty due to radar range resolution, and uncertainty due to interpolation errors (MacGregor et al., 2015). Layer depth uncertainty for the 11.7 ka isochrone is mostly below 30 m in the South-East and below 60 m in the

North-West of Greenland, for the 29 ka isochrone mostly below 40 m, for the 57 ka isochrone mostly below 50 m, and for the 115 ka isochrone mostly below 100 m (MacGregor et al., 2015).

Cross sections through the present-day ice sheet provide an intuitive way to compare modeled and observed isochrone depth for multiple isochrones at once. For the Yelmo parameter choice in the CTRL run, these cross sections show that, overall, modeled isochrones match the observed isochrones reasonably well, particularly in the western and central part of the ice sheet

(Fig. 2a–d). In the eastern part of the ice sheet the three younger modeled isochrones tend to be too high, while the 115 ka isochrone is consistently too low throughout the entire ice sheet.



Alternatively, layer depth differences can be computed for specific isochrones (Fig. 2e–h), which reveals that modeled isochrones tend to be too high in the southern parts of the ice sheet and at the eastern and western margins. Modeled isochrones depths are mostly within 400 m compared to the observed ones, and often even within 200 m. While the modeled 115 ka isochrone is depicted as too low in the panels showing cross sections (Fig. 2a–d), the layer depth difference shows less than 100 m difference (Fig. 2h). This is because modeled isochrone depth is based on the height from the bedrock upwards for the cross sections, while layer depths are computed from the ice sheet surface downwards and the different ice thicknesses of the modeled and observed ice sheets are part of the calculation (see also section 3.1).

Taking OIB uncertainty into account, the difference of modeled and observed isochrone depth normalized with OIB layer depth uncertainty provides a third perspective (Fig. 2i–l). Values for normalized isochrone depth difference range from less than –10 to more than 10. A value between –1 and 1 would be ideal, where the modeled and observed isochrone depth difference is less than the OIB isochrone uncertainty.

Both the parametrization of the ice dynamics as well as the climatic boundary conditions (mass balance) can be underlying causes for the layer depth difference patterns. The isochrone depth discrepancy in the north-eastern part is likely caused by too little precipitation during the Holocene causes (Tabone 2023, personal communication). Basal friction parameterization, which is still a large source of uncertainty in any ice sheet model (e.g., Brondex et al., 2019; Choi et al., 2022), influences ice flow throughout the simulation and can therefore contribute to model-observation mismatch in all layers.

Note that the comparison with the OIB data relies on the ability of Yelmo to simulate the ice sheet well over the last glacial cycle. Our goal in this study is not to evaluate this fit, but rather to present the capabilities of ELSA in adding accurate layer tracking to such a model. For the remainder of the study, the above CTRL run can be assumed to be the target for ELSA.

## 3.1 Limitations

The Library of Iterative Solvers for linear systems, used for solving the advection equation, shows occasional instability where layer thickness can become unrealistically large during one advection step. This error only occurs occasionally and for few isolated grid cells, often at the ice sheet boundaries where velocities are large. It is countered by adding a criterion of maximum allowed layer thickness change per advection step (see Appendix B).

Discrepancies between the BedMachine v3.1 bedrock and true bedrock cause a spiky ice sheet surface when projecting OIB ice sheet thickness onto the modeled bedrock (Fig. 2b–c, dashed dark blue line particularly on the Eastern part). Additionally, the vertical position of the isochrones is affected when visualized from the bedrock upwards, as discussed in section 3.

## 4 ELSA's results for decreased temporal, vertical, and horizontal resolution

We tested numerical strategies in ELSA to decrease run time and present how this affects the results. In the following sections, we show the dependence of isochrone depth on the choice of vertical, horizontal, and temporal resolution in ELSA and determine the error introduced compared to the CTRL run described above. To show the horizontal patterns of error magnitudes, we compare layer depths between a simulation with decreased resolution and CTRL for two specific layers: the 29 ka isochrone





and 115 ka isochrone. To display the relationship of error magnitude and isochrone age (i.e. depth in the ice sheet), we show
the RMSE computed over individual isochrones every 10 ka of layer age.

### 4.1  Decreasing the vertical resolution in ELSA

A straightforward way to decrease ELSA's computational cost is by decreasing its vertical resolution; i.e. new layers are
added less frequently and layers are thus thicker. In the CTRL setup, Yelmo is running with 10 zeta levels, which are linearly
interpolated in the vertical to provide horizontal velocities to ELSA at the required layers in the ice sheet. ELSA's default
vertical resolution is 200 a, thus much finer than the original Yelmo grid for paleo simulations of several tens of thousands of
years.

For decreased vertical resolution in ELSA, layer depth differences are positive over the majority of the ice sheet and generally
small: less than 2 m for a layer resolution of 500 a instead of 200 a (Fig. 3a–b) and between 5 m and 20 m for a layer resolution
of 2000 a instead of 200 a (Fig. 3c–d). These differences occur due to the interpolation of Yelmo velocities onto the finer vertical
grid of ELSA, and the movement of the isochrones through vertical velocity profile over the simulation. In the southern part of
the ice sheet, where surface mass balance is much larger and deeper layers are consequently much thinner, these interpolation
effects show up as dramatic differences in layer depth among neighboring grid cells and downstream effects in the direction of
flow. For comparisons with OIB data, only the 11.7 ka isochrone is affected as the older OIB layers do not cover the southern
area of Greenland.

Instead of setting a fixed vertical resolution and thus a regular vertical grid (in age) for ELSA, the desired isochrone ages can
be set manually using the parameter *age_elsa_iso* to e.g. to match the isochrone ages of the OIB data. This is the most effective
way to decrease computational cost (see Sec. 4.4). The simulation with customized isochrone ages shows similar results to
the previous experiment: layer depth errors are mostly positive and within 20 m (Fig. 3e–h). The LIS instability effect in the
southern part of the ice sheet is more pronounced.

The RMSE computed for several isochrones throughout the ice sheet ranges from less than 0.5 m for *LR*=400 a to about 4 m
for *LR*=2000 a (Fig. 4). The overall error introduced by decreasing the layer resolution is much smaller the magnitude of OIB
isochrone depth uncertainty, and reducing the vertical resolution in ELSA is an good option to decrease simulation computing
time.

### 4.2  Decreasing the horizontal resolution in ELSA

We decreased the horizontal resolution of ELSA by averaging over an integer number of host model grid cells. We varied *GF*
from 1 (default, 16 km×16 km grid) to 4 (64×64 km grid). As a result, ice thickness and isochrone depth in an ELSA grid box
show less horizontal variability compared to ELSA output from the original host model grid (Fig. 5a–d). This effect is much
greater at locations with a steeper ice thickness or isochrone depth gradient, i.e. at the ice sheet margins, with larger values of
*GF* increasing the magnitude of this effect. The overall error pattern does not seem to be related to mass balance or dynamics
of the ice sheets.



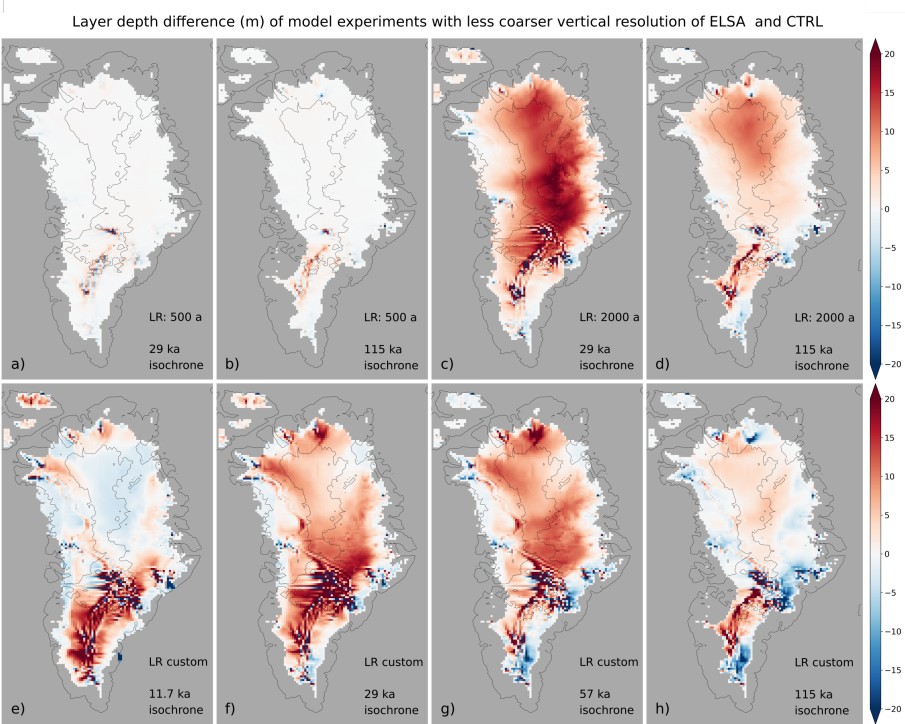

**Figure 3.** Layer depth difference (m) of model experiments with a coarser ELSA vertical grid and CTRL: *LR*=500 a for a) the 29 ka isochrone and b) the 115 ka isochrone; *LR*=2000 a for c) the 29 ka isochrone and d) the 115 ka isochrone. The bottom row shows layer depth difference between a simulation with customized isochrone ages and CTRL for the isochrones that match the OIB isochrone ages.

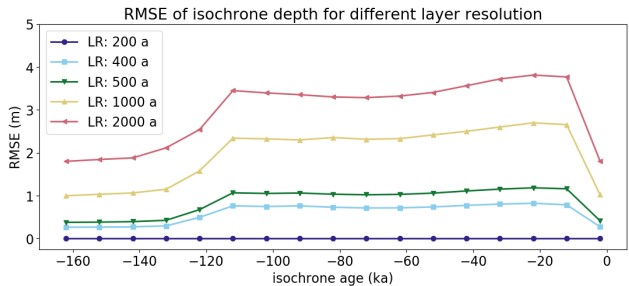

**Figure 4.** RMSE computed for layer depth difference between model experiments with a coarser vertical ELSA grid and CTRL.

In the ice sheet center, the errors range from -50 m to 40 m for *GF*=2 (Fig. 5a–b) and from -120 m to 50 m for *GF*=4 (Fig. 5c–d). At the margins, the errors reach values of more than 500 m. Linearly interpolating the final results from the coarser grid back to the original grid particularly smooths the error in the center of the ice sheet and reduces them to the values around 30 m (Fig. 5e–h). The errors at the margins are mostly unaffected by the interpolation and remain large.





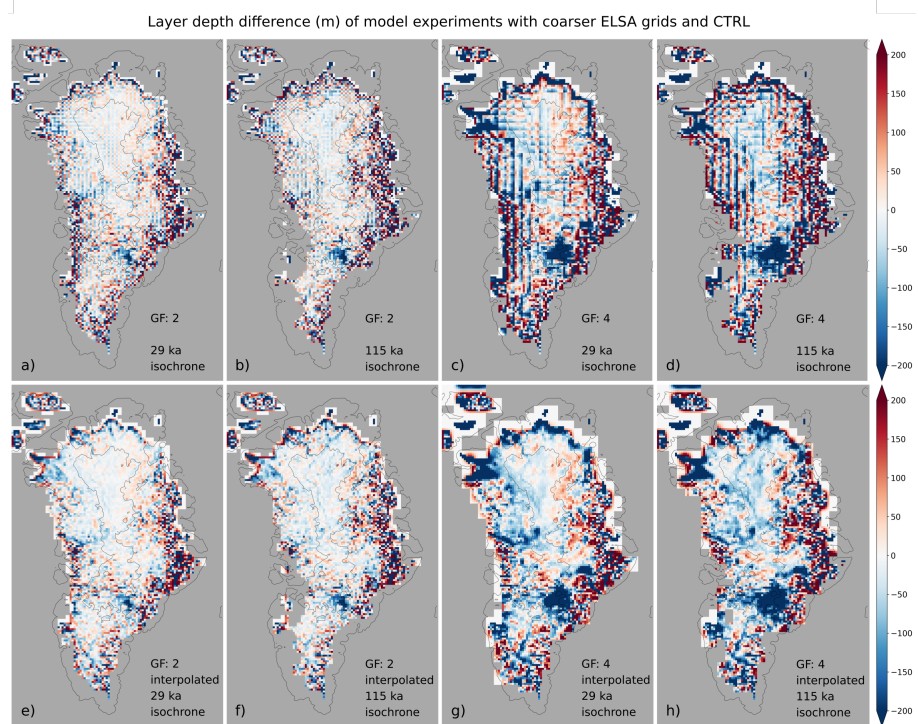

**Figure 5.** Layer depth difference (m) of model experiments with a coarser ELSA horizontal grid and CTRL: *GF*=2 for a) the 29 ka isochrone and b) the 115 ka isochrone; *GF*=4 for c) the 29 ka isochrone and d) the 115 ka isochrone. The bottom row shows results for the same settings as the top row, but with ELSA output linearly interpolated back to the original grid.

In summary, the overall error introduced by increasing *GF* is about 0.5–3 times the magnitude of OIB isochrone depth uncertainty, depending on the location. The larger errors at the margins for increased *GF* are not that relevant to isochrone applications as the radiostratigraphy data are only available for the ice sheet center and the issues at the margins do not backpropagate to the ice sheet center.

The RMSE computed for several isochrones throughout the ice sheet ranges from about 120 m for *GF*=2 to 200 m for *GF*=4 (Fig. 6, solid lines). These large RMSEs are strongly influenced by large differences at the ice sheet margins between the *GF* experiments and CTRL. Over isochrone age, the RMSE stays mostly constant. The exception are the youngest few layers, where RMSE reduces by about 25 % due to smaller differences at the ice sheet margins, particularly in the North-East (not shown).

Interpolation to the original grid reduces the RMSE in to about 100 m for *GF*=2 and to about 160 m for *GF*=4 (Fig. 6, dashed lines). Since the interpolation mainly smooths differences in the ice sheet center, but does not improve the results at the margins much, the RMSE remains high overall.



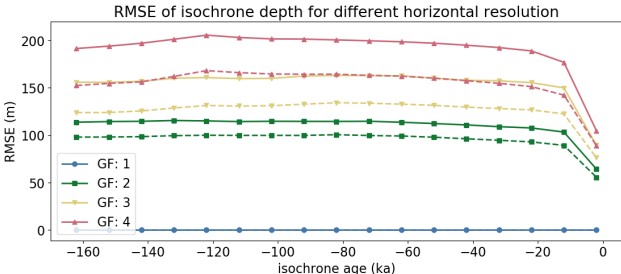

**Figure 6.** RMSE computed for layer depth difference between model experiments with coarser ELSA horizontal grid and CTRL. Solid lines are the RMSEs on the resulting coarser grid, dashed lines are the RMSEs for linear interpolation back to the original grid.

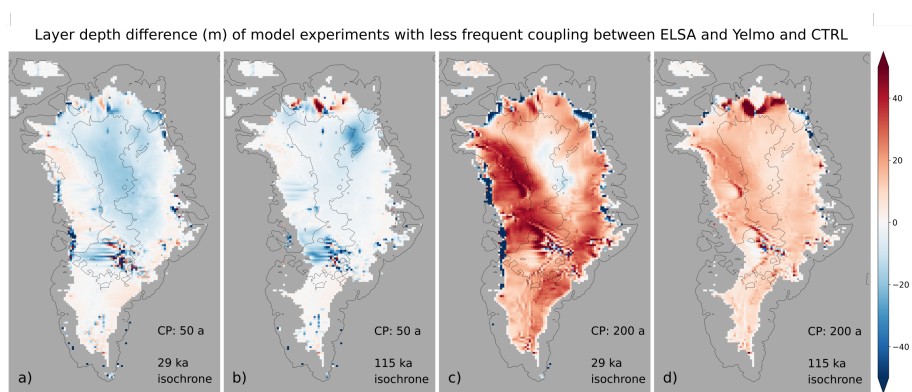

**Figure 7.** Layer depth difference (m) of model experiments with less frequent coupling of ELSA and Yelmo: $CP$=50 a for a) the 29 ka isochrone and b) the 115 ka isochrone; $CP$=200 a for c) the 29 ka isochrone and d) the 115 ka isochrone.

## 4.3 Decreasing the temporal resolution of ELSA

The temporal resolution of ELSA can be reduced by increasing the coupling period $CP$ between the host model and ELSA, as it would also be the case in an offline application of ELSA using saved host model output. We varied $CP$ from 10 a (default) to

200 a ($UF$ from 1 to 20).

Layer depth differences for a coupling period of 50 a range from –25 m to 5 m (Fig. 7a–b), while differences for a coupling period of 200 a are primarily positive and mostly range from 5 m to 45 m Fig. 7c–d). These errors are smaller than or comparable to OIB layer depth uncertainty.

The RMSE computed for several isochrones throughout the ice sheet depicts the instability of varying coupling periods: For

values of $CP$ except for $CP$=40 a, the RMSE is below 40 m and is close to constant with isochrone age. For $CP$=40 a, however, the RMSE increases up to 80 m and the two dimensional layer depth difference over specific isochrones is very irregular (not shown).





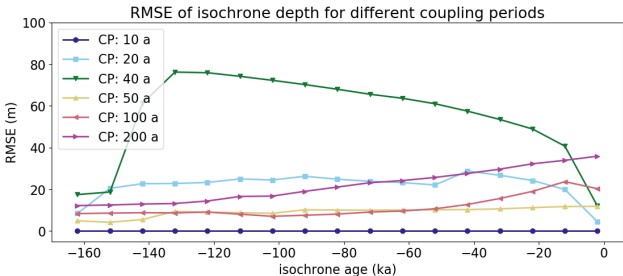

**Figure 8.** RMSE computed for layer depth difference between model experiments with longer coupling periods ELSA and Yelmo and CTRL.

This model instability is relevant when using ELSA offline and forcing it with ice sheet model output if the temporal resolution of the saved host model output is not sufficiently high. Further investigation of offline setup with specific host models is recommended.

### 4.4 Resolution and model speed

With the right setup, it is computationally quite cheap to add ELSA to an ice sheet model run, with the added benefit of enabling the evaluation of the run using ELSA's output and OIB data. Computational cost is evaluated through model run speed (kiloyear simulation time per computational hour). The run speed of Yelmo itself is $20.8\,\mathrm{ka\,h^{-1}}$ (Fig. 9, black dashed line) in our single thread setup. When customized to only 13 specific isochrones, the coupled ELSA-Yelmo model setup runs with $18.9\,\mathrm{ka\,h^{-1}}$, almost as fast as the Yelmo stand-alone speed. Run speed drops to $4.4\,\mathrm{ka\,h^{-1}}$ for the coupled ELSA-Yelmo setup with the CTRL parameter settings (dark blue bars). Decreasing the vertical, horizontal, and temporal resolution increases run speed, but gain in run speed is not linearly proportional to the decrease in resolution but logarithmic; furthermore, a decrease in horizontal and temporal resolution introduces errors.

Picking a few specific isochrones of interest is therefore by far the fastest and most accurate coupled model run. Adding isochrones on a regular vertical grid may be necessary for runs on shorter time scales to investigate flow patterns within the ice sheet. Reduced temporal resolution is most likely necessary when using ELSA offline and host model output is not available at every model time step.

## 5 Conclusions

We have described the workflow and features of Englacial Layer Simulation Architecture (ELSA), an isochronal model that can be coupled to a full ice-sheet model (host model) for layer age tracing in an ice sheet's interior. ELSA models individual layers of accumulation explicitly, driven by surface mass balance. Each layer added is isochronal, i.e. has a fixed time stamp. As time passes and more layers accumulate, the ice of older layers flows towards the margin of the ice sheet and the layers become thinner. This straightforward approach to model the englacial stratigraphy avoids issues present in alternative layer age tracers, such as numerical diffusion.



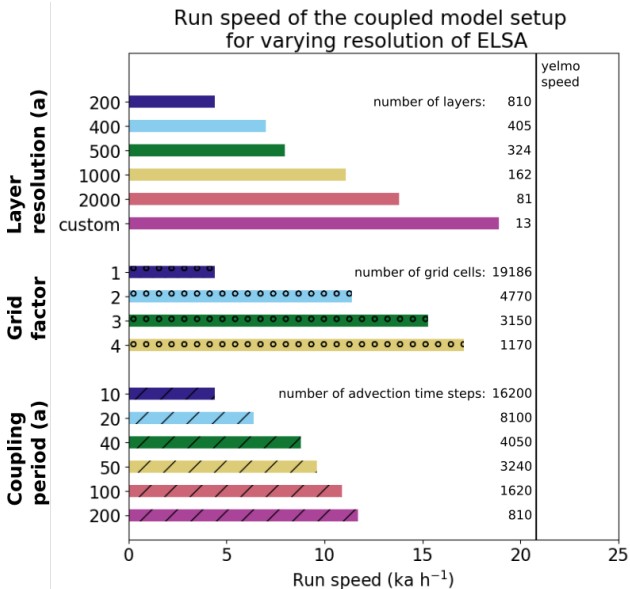

**Figure 9.** Run speed ($\text{ka h}^{-1}$) for different parametrizations of ELSA's vertical (top bars), horizontal (middle bars), and temporal (bottom bars) resolution, as used in the experiments in section 4. The black solid line shows run speed for the Yelmo stand-alone run. The greatest increase in run speed is achieved when choosing a few specific isochrones of interest. Initialization layers are not included in the number of layers noted in the figure.

ELSA's computational cost can be decreased by reducing its vertical, horizontal, or temporal resolution. The fastest and preferred way is specifying a number of isochrones instead of a regular vertical grid, which not only increases run speed, but also introduces only a very small error. Decreasing the horizontal resolution of ELSA with respect to the host model also leads to an increase in run speed, but introduces errors of the order of tens to hundreds of meters, depending on the degree of resolution depreciation and location. Decreasing the temporal resolution of ELSA with respect to the host model (i.e. less frequent coupling) introduces an error magnitude comparable to OIB uncertainty; however, the model runs are not stable in all configurations.

To our knowledge, this is currently the only isochronal model for ice sheets and will thus be a valuable comparison tool with Eulerian, Lagrangian, and Semi-Lagrangian layer tracers. Applicability also includes the usage of ELSA for a host model's parameter tuning via ensemble runs (Born and Robinson, 2021). Lastly, this work is the foundation for an offline application of ELSA, further increasing opportunities to evaluate any given ice-sheet model based on its englacial stratigraphy.

## Appendix A: Control run results

Figure A1 shows the Yelmo results of the CTRL run (described in Section 3): ice thickness and extent at the end of the simulation (year 2000 AD), ice thickness and extent during the last glacial maximum, ice thickness and extent during the





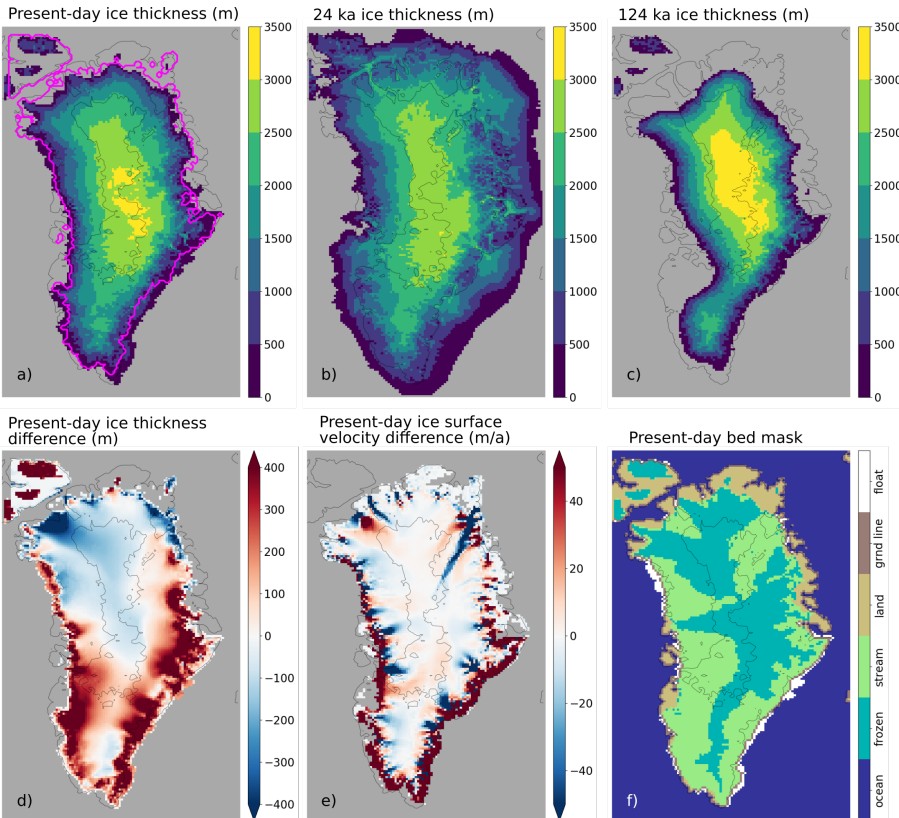

**Figure A1.** Model CTRL run results: a) Present-day ice thickness (pink line marks the BedMachine v3.1 outline (Morlighem et al., 2017)), b) ice thickness and extent during the last glacial maximum, c) ice thickness and extent during the Eemian, d) present-day ice thickness difference compared to BedMachine v3.1, e) present-day velocity difference compared to Joughin et al. (2018), f) present-day bed properties.

Eemian, the difference of present-day ice thickness wrt. BedMachine v3.1, the difference of present-day ice surface velocity wrt. to Joughin et al. (2018), and bed properties at the end of the simulation.

## Appendix B: Layer advection

ELSA uses a Lagrangian description of flow in the vertical dimension, but the computationally cheaper Eulerian description of flow in the horizontal dimension within the individual layers. With this approach, numerical diffusion is eliminated by design in the vertical as layers never exchange mass. In the horizontal, numerical diffusion does affect the flow as it uses a finite-difference scheme. However, numerical diffusion comes about in the presence of steep gradients, and natural tracers in ice sheets generally vary smoothly in the horizontal (see also Born, 2017).



The change of the layer thickness $d$ in $x$ and $y$ direction over time $t$ is given by the divergence of the flow of volume $F$:

$$\frac{\partial d}{\partial t} = -\nabla \cdot \mathbf{F} = -\left(\frac{\partial}{\partial x}, \frac{\partial}{\partial y}\right) \cdot \mathbf{F}(x,y) = -\frac{\partial}{\partial x} ud - \frac{\partial}{\partial y} vd \tag{B1}$$

where $u$ and $v$ are the horizontal ice velocities in $x$ and $y$ direction, respectively.

Equation B1 is discretized forward in time and upstream in space to

$$\frac{d_{i,j}^{t+1} - d_{i,j}^{t}}{\Delta t} = -u_{i-1/2,j}^{t} \cdot \frac{d_{i,j}^{t+1} - d_{i-1,j}^{t+1}}{\Delta x} - d_{i,j}^{t+1} \cdot \frac{u_{i+1/2,j}^{t} - u_{i-1/2,j}^{t}}{\Delta x} - v_{i,j-1/2}^{t} \cdot \frac{d_{i,j}^{t+1} - d_{i,j-1}^{t+1}}{\Delta y} - d_{i,j}^{t+1} \cdot \frac{v_{i,j+1/2}^{t} - v_{i,j-1/2}^{t}}{\Delta y} \tag{B2}$$

for positive values of $u$ and $v$ and

$$\frac{d_{i,j}^{t+1} - d_{i,j}^{t}}{\Delta t} = -u_{i+1/2,j}^{t} \cdot \frac{d_{i+1,j}^{t+1} - d_{i,j}^{t+1}}{\Delta x} - d_{i,j}^{t+1} \cdot \frac{u_{i+1/2,j}^{t} - u_{i-1/2,j}^{t}}{\Delta x} - v_{i,j+1/2}^{t} \cdot \frac{d_{i,j+1}^{t+1} - d_{i,j}^{t+1}}{\Delta y} - d_{i,j}^{t+1} \cdot \frac{v_{i,j+1/2}^{t} - v_{i,j-1/2}^{t}}{\Delta y} \tag{B3}$$

for negative values of $u$ and $v$. Velocities are used at the faces of the grid cells.

We solve implicitly for $d^{t+1}$ with

$$\boldsymbol{d^t} = \mathbf{A}\boldsymbol{d^{t+1}} \tag{B4}$$

where the main and secondary diagonals of $\mathbf{A}$ are filled using Eqs. B2 and B3 and are given as

$$d_{i,j}^{t} = d_{i,j}^{t+1}\left(1 + \frac{\Delta t}{\Delta x}u_{i+1/2,j}^{t} + \frac{\Delta t}{\Delta y}v_{i+1/2,j}^{t}\right) - d_{i-1,j}^{t+1}\frac{\Delta t}{\Delta x}u_{i-1/2,j}^{t} - d_{i,j-1}^{t+1}\frac{\Delta t}{\Delta y}v_{i-1/2,j}^{t} \qquad \text{for } u_{i,j} > 0.1, v_{i,j} > 0.1 \tag{B5}$$


$$d_{i,j}^{t} = d_{i,j}^{t+1}\left(1 - \frac{\Delta t}{\Delta x}u_{i-1/2,j}^{t} - \frac{\Delta t}{\Delta y}v_{i-1/2,j}^{t}\right) + d_{i+1,j}^{t+1}\frac{\Delta t}{\Delta x}u_{i+1/2,j}^{t} + d_{i,j+1}^{t+1}\frac{\Delta t}{\Delta y}v_{i+1/2,j}^{t} \qquad \text{for } u_{i,j} < -0.1, v_{i,j} < -0.1 \tag{B6}$$

$$d_{i,j}^{t} = d_{i,j}^{t+1}\left(1 + \frac{\Delta t}{\Delta x}u_{i+1/2,j}^{t} - \frac{\Delta t}{\Delta y}v_{i-1/2,j}^{t}\right) - d_{i-1,j}^{t+1}\frac{\Delta t}{\Delta x}u_{i-1/2,j}^{t} + d_{i,j+1}^{t+1}\frac{\Delta t}{\Delta y}v_{i+1/2,j}^{t} \qquad \text{for } u_{i,j} > 0.1, v_{i,j} < -0.1 \tag{B7}$$

$$d_{i,j}^{t} = d_{i,j}^{t+1}\left(1 - \frac{\Delta t}{\Delta x}u_{i-1/2,j}^{t} + \frac{\Delta t}{\Delta y}v_{i+1/2,j}^{t}\right) + d_{i+1,j}^{t+1}\frac{\Delta t}{\Delta x}u_{i+1/2,j}^{t} - d_{i,j+1}^{t+1}\frac{\Delta t}{\Delta y}v_{i-1/2,j}^{t} \qquad \text{for } u_{i,j} < -0.1, v_{i,j} > 0.1 \tag{B8}$$

At the domain boundaries as well as for $-0.1 < u < 0.1\,\mathrm{ma}^{-1}$ and $-0.1 < v < 0.1\,\mathrm{ma}^{-1}$, layer thickness is unchanged.

The Library of Iterative Solvers for linear systems shows occasional instabilities where layer thickness become unrealistically large during one advection step. We introduced a maximum allowed layer thickness change per advection step, $d_{\mathrm{diff}}$ (m), which is defined as:

$$d_{\max} = 100 + 10\,\text{update\_factor} + \text{layer\_resolution}/10 \tag{B9}$$

For the default settings of update_factor=1 and layer_resolution=200, $d_{\mathrm{diff}}$=130 m.



*Code availability.* ELSA's source code is available at https://git.app.uib.no/melt-team-bergen/elsa under the license GPL-3.0, including instructions for the ELSA and Yelmo coupling. The version for this manuscript is the tagged version v2.0. For the parametrization of the CTRL run, please access the namelist file GRL16_PMIP-2-pal_0001.nml at

https://git.app.uib.no/melt-team-bergen/elsa/-/blob/main/model_integration/yelmox_v1.801/GRL16_PMIP-2-pal_0001.nml?ref_type=heads. The source code of Yelmo and Yelmox is available at https://github.com/palma-ice/yelmo and https://github.com/palma-ice/yelmox under the license GPL-3.0. We used version 1.801 (tag v1.801) for both Yelmo and Yelmox for this manuscript.

*Author contributions.* TR designed the study with contributions from AB and AR. TR ran all experiments, performed the analysis, and wrote the manuscript with input from AB. AR provided support regarding the Yelmo setup, input data, and structure of the study. AB, TR, and RL

wrote and published the ELSA code. GG contributed to the data evaluation and plotting code.

*Competing interests.* The authors declare that they have no conflict of interest.

*Acknowledgements.* TR, AB, and RL received funding from the Norwegian Research Council Grant 314614 (Simulating Ice Cores and Englacial Tracers in the Greenland Ice Sheet). AR received funding from the European Research Council Grant 101044247 (FORCLIMA). GG was supported by the German Academic Exchange Service. We sincerely thank Ilaria Tabone, who provided advice on the parametriza-

tion of Yelmo, and Philipp Voigt for his input on the ELSA code.



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
