# Peer review of "Design and performance of ELSA v2.0: an isochronal model for ice-sheet layer tracing"

_EGUsphere, 2023_

## Author Response (AR1)

We thank both reviewers for their comments and suggestions. We are responding to all points below, the reviewer's comments are in italic font.

**REVIEWER 1:**

*The novelty compared to (Rybak and Huybrechts, 2003) is that the whole isochronous surface is tracked, not individual particles.*

*Though the authors stress (lines 184-185) that their main goal is not to reach as much closer fit to what they call "observations", but rather to present capabilities of the method, I would suggest to enlarge the section 3.1 and discuss the problem of distortion and overturning of ice layers close to bottom (NEEM community members, 2013; fig. 13 in (MacGregor et al., 2015)). In view of the severe problems close to the bottom of the ice sheet, the choice of the 115 ka isochrone as target one in the model tests looks ret very reliable and must be additionally justified.*

*For me, it is not clear how authors technically deal with the lateral ice melting and calving in terms of isochronous surfaces propagation, as well as with changes in ice sheet configuration (retreat-advance) and with basal melting. I suggest to discuss this issue in the revised manuscript.*

We agree that ice layer folds could generally present an issue. This is a problem related to depth as it is to age, and this relationship is not the same across the ice sheet. Furthermore, there are 2 aspects about the folds: the model and the reconstructions. Regarding the modeling aspect, we (or our host model) do not model folds, and thus have no way to specifically address them. Regarding the reconstructed data, we only use the (extensively) postprocessed radiostratigraphy dataset, which we assume is quality checked. Where the 115 ka isochrone is defined in this data, we presume that MacGregor et al. (2016) found the effect of folding to be negligible. Furthermore, the gridded part of the dated radiostratigraphy dataset defines the 115 ka isochrone only in the north, where accumulation is lower and the 115 ka isochrone is thus closer to the surface. Our conclusion is to still use the 115 ka isochrone as a target, but be mindful of the limitations and not to give it too much weight in the interpretation. Going into detail of isochrones modeling with respect to folds and basal processes is out of the scope of this paper.

We have added the following paragraph on Page 7 line 166: "Note that folding of layers at the base of the ice sheet can present an issue. Layer folding is not modeled by Yelmo or ELSA. Since the reconstructed data we use is processed and quality controlled, we assume that where the 115 ka isochrone is defined by MacGregor et al. (2015), the presence of folding disturbances has not notably influenced the isochrone's large-scale characteristics. Furthermore, the gridded part of the dated radiostratigraphy dataset defines the 115 ka isochrone only in the north, where accumulation is lower and the 115 ka isochrone is thus further away from the base of the ice sheet and closer to the surface. Nevertheless, the specific

shortcomings of the 115 ka isochrone should be kept in mind when using it for model-reconstruction comparison."

Lateral advection in ELSA is in accordance with lateral advection in the host model. Lateral ice melting and calving is dealt with in the host model. The layers in ELSA are normalized using the host model's ice thickness.

Regarding retreat-advance, ELSA is using the input from the host model.

Basal melting is discussed in lines 89-92. Over the course of the simulation of an entire glacial cycle, basal melt was not enough to deplete the initialization layers.

*Particular notes*

*Line 16: I suggest to add a reference to (EPICA community members, 2006).*

We have added the reference.

*Line 16: "ice core data are limited to specific locations". This is not exactly correct. Because of ice flow, ice core data characterize climate change in the past on a relatively big territory – the deeper is the layer in the ice core the more remote is the place of origin of the ice (see eg. (Huybrechts et. al, 2007 in Climate of the Past; EPICA Community members, 2006, in Nature – specifically, Supplementary materials; NEEM community members, 2013, in Nature; Huybrechts et al., 2009, in Annals of Glaciology). I suggest to add a paragraph discussing the problem.*

We agree that our original wording was imprecise. However, we do not consider it necessary to add a full paragraph discussing this problem since it only loosely connects to our main issue addressed in this manuscript - modeling englacial layers. We have edited the sentence in line 16 to "However, ice core data are sparse due to the high cost and effort associated with their retrieval, and therefore they represent only a very limited spatial scale on large ice sheets."

*Line 57: The description is somewhat vague – how can the vertical axis be defined in time RATHER than in space? Either in time or in space, I think, and not a little bit here and a little bit there. In the previous papers (Born, 2017; Born and Robinson, 2021) this issue is enlightened rather clear. I suggest to describe vertical discretization in the more clear way as well as propagation of the isochrone surfaces (layers) in the horizontal (in the vertical, too) in the more clear way.*

We rephrased the sentence to: "While the vertical grid of ice sheet models is commonly defined in space, ELSA's vertical grid is defined in time (Born, 2017)."

The horizontal propagation is described in lines 92-95. We expanded this section in the revised manuscript.

We added information in line 54 to better describe the vertical propagation: "Layers never exchange mass, they only become thinner as ice flows towards the margin of the ice sheet and calves or melts."

*Line 134: Authors use Shapiro and Ritzwoller (2004) geothermal heat flux (GHF) field to calculate basal melting. Though the GHF field in the cited work seems to correctly reproduce the reality in general, I am not sure about the North-Western Greenland around stations NEEM and especially NorthGRIP, where the bottom layers older than 120-124 ka BP disappeared because of relatively high basal melting rate probably caused by locally enhanced GHF.*

There is a lot of uncertainty attached to the GHF field and no recommendation or consistent use, even for simulations for the Ice Sheet Model Intercomparison Project for CMIP6 simulations (Zhang, 2023) . Since the Shapiro and Ritzwoller (2004) flux is commonly used and our main goal is to highlight the capabilities and limitations of ELSA in this manuscript, our main concern is not the best possible parameterization of the host model and we therefore consider the Shapiro and Ritzwoller (2004) GHF field appropriate for our purposes.

*Caption to Fig. 2. Authors use term "observed" for the OIB data which is confusing. In lines 151-162 authors describe how the OIB chronology was derived as a combination of radiostratigraphy and ice core dating (which is model derived in deeper layers).*

Thank you, this is a good point. We changed "observed" to "reconstructed" in this sentence, and throughout the manuscript when referring to OIB isochrones.

*Figure 2, panels a-c: the difference in depth between modelled and the OIB isochrones is confusing, especially if we consider not very "old" ages – 11,7 and 29 ka BB isochrones. These results must be explained and discussed, because the discrepancy in estimates may question the possibility of practical implementation of the method. In this view, I do not agree with the authors (Lines 162-166) that the model reproduces radiostratigraphy "reasonably well". For instance, in panel b, isochrones 11,7 ka ("observed") and 29 ka (modelled) merge. Same for 29 ka and 57 ka. I cannot qualify this result as "reasonably well" even taking into account uncertainty in "observed" values.*

We agree that there is a mismatch for reconstructed and modeled isochrones, as we would expect given that the focus of this manuscript is not finding the parametrization for a best simulation over the last glacial cycle. We spent some time to find a parametrization of the host model that arrives reasonably close to present-day Greenland ice sheet thickness and extent, as well as bed properties (frozen vs thawed). Setting up a large ensemble for an improved host model control run is out of the scope of this work. The discrepancy in reconstructed and modeled isochrones does not affect the implementation of ELSA — this is purely a host model parameterization issue. It also *exactly* presents the usefulness of ELSA for host model tuning — the present-day ice sheet looks reasonable, as well as a number of other qualifying parameters (presented in Appendix A), but the internal layer structure shows that the prescribed ice dynamics, surface mass balance, and/or boundary conditions still produce a mismatch for

modded vs reconstructed isochrones over the course of the simulation.

**REVIEWER 2:**

*This paper concerns what I'd describe as a performance tradeoff analysis of a presently unique model (ELSA) that cleverly couples offline with a full ice-sheet model (Yelmo) to generate the resulting synthetic age structure of the ice sheet. The model described is appealingly simple and so the number of model parameters is small, but they are all fairly considered in terms of their impact on the output. The result is a compact study that mostly stands on its merits but could use better context with what has preceded it. I have mostly glaciological and non-modeler type concerns about the paper as it stands. I've noted one bigger issue on which that I think the paper needs substantially more clarity for its intended audience.*

*Major concern:*

*It took a careful reading and checking of references for me to understand (or think I do) how ELSA vertically advects the age structure and thins the layers at each time step. At first, I was surprised not to see any mention of \*vertical\* velocities in Figure 1. I wondered if ELSA used a Nye sandwich model throughout the ice sheet, which would not be good. Then I realized that the vertical velocities must simply be determined by continuity from the host model's 3-D horizontal velocity field. That's fine, but not expressed clearly in the paper. However, I then became curious about the nature of the host model itself and found that it uses a depth-averaged approximation of vertical shear in the ice column (129-130). While substantial and informative work has been done on this model's properties (Born and Robinson, 2021; Robinson et al., 2022), I find this an odd choice in the context of resolving the 3-D age structure of an ice sheet as it forces a pre-determined vertical velocity pattern (if not magnitude) on the ice sheet, which is a risk considering past observations (Fahnestock et al., 2001, 10.1126/science.1065370). A potential resolution here would be greater clarity in the model description as to what happens to the isochronal layers \*inside\* the ice sheet at each time step (not just the top and bottom), and later a more robust discussion of better ways to model an age structure generally. The latter part tends more toward the host model physics and other inputs, so maybe a summary of the conclusions of Born and Robinson (2021) is appropriate.*

Yelmo is an ice-sheet model that fully resolves the 3D velocity field, with the 3D shear-stress fully represented within the solver. To calculate the effective viscosity (a 3D field), the longitudinal strain rate terms (du/dx, dv/dy) are approximated by the 2D depth-averaged fields, which turns out to be a reasonable approach. A model like Yelmo using the DIVA velocity solver cannot resolve more complex flow like ice-folding, but can otherwise be expected to represent large-scale continental ice flow to high fidelity. These details are clarified in the revised text to avoid confusion. We added the following text on page 6 line 130: "Yelmo fully resolves the 3D velocity field, with the 3D shear-stress fully represented within the solver. To calculate the 3D field of the effective viscosity, the longitudinal strain rate terms (du/dx, dv/dy) are approximated by the 2D depth-averaged fields, which turns out to be a reasonable approach. A model like

Yelmo using the DIVA velocity solver cannot resolve more complex flow like ice-folding, but can otherwise be expected to represent large-scale continental ice flow to high fidelity."

**Note that ELSA does not use vertical velocities from the host model at all as vertical movement in ELSA strictly results from changes in the individual layers' thickness.** We have made this clearer now in the section describing the evolution of isochrones in the ice sheet in lines 92-95: "ELSA's individual layers are advected in the horizontal dimension using a Eulerian description of flow (see Appendix B). The passive tracer variable is the layer thickness d, which is advected using an implicit upstream scheme and the host model's horizontal velocities, which are linearly interpolated in the vertical from the original host model's vertical grid of the ice sheet onto the isochronal grid. All layers are advected, where advection is strictly two-dimensional within each isochrone. Therefore, vertical velocities from the host model are not required, and vertical movement in ELSA strictly results from changes in the individual layers' thickness. Layers change thickness due to advection, but they never exchange mass."

We also edited lines 57-60 that relate to this topic.

We made edits in section 3 to be clearer on distinguishing results from the host model vs ELSA. Section 3 now has a subsection 3.1 describing the host model setup, subsection 3.2 describing the ELSA setup, subsection 3.3 introducing the radiostratigraphy data set, subsection 3.4 describing the results of the Control run, and subsection 3.5 describing ELSA's limitations.

*1: Is it correct to describe this paper as "Introducing ELSA v2.0…", given the verbiage on L41? That suggests that Born and Robinson (2021) "introduced" ELSA v2.0. This paper still stands on its own merits and is certainly a more complete description. I wonder if a different title is more appropriate, e.g., "Design and performance of ELSA v2.0…".*

It is true that Born and Robinson (2021) introduced ELSA, although not under that name yet. But the manuscript describes an updated version compared to the original isochronal layer tracer, and we have changed the title to "Design and performance of ELSA v2.0: an isochronal model for ice-sheet layer tracing".

*23: Is the reference to "Standard tools" those for glaciology or a different field? More context here is appropriate before the relevant citations later on in this paragraph.*

We changed the sentence to: "Standard passive tracer tools in glaciology are Eulerian and Lagrangian tracer schemes."

*132: Clarify what a "sigma-coordinate grid" is for non-modelers. Also, how are they different from "zeta levels" mentioned later on (L203)?*

We changed the sentence in line 132 to: "The vertical grid is a sigma-coordinate grid (Greve and Blatter, 2009), where the vertical axis represents the relative height within the ice sheet from 0 at the base to 1 at the surface. Sigma surfaces follow model terrain and continuous fields are represented smoothly even in the lowest layers in the model. We use a sigma grid with 10 layers with a higher resolution at the base of the ice sheet."

We changed the sentence in line 203 to: "In the CTRL setup, Yelmo is running with 10 vertical levels, which are linearly interpolated.."

*144: Which "bed properties"? There are several relevant ones.*

We changed the sentence in line 144 to: "..and the present-day bed property state (frozen vs thawed) is similar to the one assembled by MacGregor et al. (2022)."

*148 and elsewhere: For paleoclimate in much more common usage these days is the use of Common Era (CE) instead of AD for historical ages.*

We changed AD to CE throughout the manuscript.

*151: There is also pre-OIB data used by MacGregor et al. (2015).*

We refer to MacGregor et al. (2015) instead of OIB throughout the paper now.

*178: parameterization*

Thank you, we corrected it.

*186: If there's no section 3.2, then there doesn't need to be a section 3.1. Also, this sub-section may be more appropriate for the Discussion.*

We split section 3 in several subsections in the revised manuscript. We kept the section on limitations in section 3, as it seems appropriate to discuss these limitations right away.

*191: I'm familiar with BedMachine and haven't heard of v3.1. It's not mentioned in Morlighem et al. (2017), which is simply described as v3. Regardless, BedMachine v5 has been out since mid-2022. I realize that big ice-sheet models are strangely slow at updating their boundary conditions, hence the use of Shapiro and Ritzwoller (2004…20 years ago…) for geothermal flux, but some clarity or nomenclature adjustment is needed here.*

We corrected the version to BedMachine v3 throughout the manuscript.

*204: vertical dimension*

We corrected the sentence to: "..which are linearly interpolated in the vertical dimension.."

*263: I'm not knowledgeable in these matters or what GMD policy is, but more commonly whenever runtimes are mentioned or compared I've also seen the processor / # cores / etc. described.*

We added the following information in line 263: "Computational cost is evaluated through model run speed (kiloyear simulation time per computational hour) on a compute node using Intel Xeon Gold 6136 CPUs."

*Figure 2e-l/3/5/7/A1: Could some colors be added other than gray to distinguish between the ocean, land and ice where the isochrones are not mapped? It's a bit drab as is with just the present-day coastline.*

We chose to leave the figures as they were as more colors do not add any important information and, in our opinion, is distracting from the relevant information in the figures.

---

## Referee Report (RR1)

The revised version of the manuscript looks better compared to the initially submitted one. Still, I am not convinced with the argumentation concerning mismatches between modelled and reconstructed isochrone depths in Fig. 2. It is obvious that the model fails to reproduce reconstructed isochrone depth in some regions. Of course, one can suggest possible explanations what is the reason for that. But without due proof that the mismatch was caused by e.g. insufficient climatic forcing, one can think about failures in the modelling approach. Anyway, I realise that the main goal of the paper was to demonstrate a new methodology in the model-based paleo-reconstructions, though the demonstration of the confusing results decreases the value of the approach.

---

## Author Response (AR2)

We thank both reviewers and the editor for their evaluation of the manuscript.

We have corrected the typos for the production file upload.